# Change in Secondary Metabolites and Expression Pattern of Key Rosmarinic Acid Related Genes in Iranian Lemon Balm (*Melissa officinalis* L.) Ecotypes Using Methyl Jasmonate Treatments

**DOI:** 10.3390/molecules27051715

**Published:** 2022-03-06

**Authors:** Farzad Kianersi, Davood Amin Azarm, Alireza Pour-Aboughadareh, Peter Poczai

**Affiliations:** 1Department of Agronomy and Plant Breeding, Faculty of Agriculture, Bu-Ali Sina University, Hamedan P.O. Box 6517838695, Iran; 2Department of Horticulture Crop Research, Isfahan Agricultural and Natural Resources Research and Education Center, AREEO, Isfahan P.O. Box 81785199, Iran; afiunidavood@gmail.com; 3Seed and Plant Improvement Institute, Agricultural Research, Education and Extension Organization (AREEO), Karaj P.O. Box 3158854119, Iran; a.poraboghadareh@edu.ikiu.ac.ir; 4Botany Unit, Finnish Museum of Natural History, University of Helsinki, P.O. Box 7, FI-00014 Helsinki, Finland

**Keywords:** lemon balm, rosmarinic acid, methyl jasmonate, gene expression, total phenolic, total flavonoid content

## Abstract

The medicinal herb, lemon balm (*Melissa officinalis* L.), which is high in rosmarinic acid (RA), has well-known therapeutic value. The goals of this study were to investigate the effects of methyl jasmonate (MeJA) on RA content, total phenolic content (TPC), and total flavonoid content (TFC), as well as changes in expression of their biosynthesis-related key genes (*MoPAL*, *Mo4CL*, and *MoRAS*) in Iranian lemon balm ecotypes, as first reported. Our results revealed that MeJA doses significantly increase the RA content, TPC, and TFC in both ecotypes compared with the control samples. Additionally, the higher expression levels of *MoPAL*, *Mo4CL*, and *MoRAS* following treatment were linked to RA accumulation in all treatments for both Iranian lemon balm ecotypes. After 24 h of exposure to 150 µM MeJA concentration, HPLC analysis showed that MeJA significantly increased RA content in Esfahan and Ilam ecotypes, which was about 4.18- and 7.43-fold higher than untreated plants. Our findings suggested that MeJA has a considerable influence on RA, TPC, and TFC accumulation in MeJA-treated Iranian *M. officinalis*, which might be the result of gene activation from the phenylpropanoid pathway. As a result of our findings, we now have a better understanding of the molecular processes behind RA production in lemon balm plants.

## 1. Introduction

Lemon balm (*Melissa officinalis* L.) is a common medicinal herb in the Lamiaceae family. Its sedative, carminative, spasmolytic, antibacterial, and antiviral activities are attributed to the presence of essential oils (citral, citronella) and phenolic chemicals [1]. The primary phenylpropanoid component in the medicinal plant *M. officinalis* has been identified as rosmarinic acid (RA) [2]. Antioxidant, anti-inflammatory, and antibacterial properties are all present in this molecule [3,4,5]. RA, the dominant active phenolic compound in *M. officinalis*, is an ester of caffeic acid and 3,4-dihydroxyphenyllactic acid. It is a naturally occurring chemical found in a variety of therapeutic plants and herbs [6]. RA was shown to be an active component in various medicinal plants belonging to the Lamiaceae family, but it may also be found in plants from all over the world [7,8].

As part of the phenolic acid biosynthesis pathway, the amino acids L-phenylalanine and L-tyrosine are used as precursors to manufacture RA (Figure 1) [9]. L-phenylalanine is used to make caffeic acid, whereas L-tyrosine is used to make 3,4-dihydroxyphenyllactic acid [10]. Phenylalanine ammonia-lyase (PAL; EC 4.3.1.24), cinnamic acid 4-hydroxylase (C4H; EC 1.14.14.91), and coumaric acid CoA ligase are the enzymes involved in the conversion of L-phenylalanine to para-coumaroyl-coenzyme A (CoA) precursor (4CL; EC 6.2.1.12). PAL deaminates L-phenylalanine oxidatively, then C4H hydroxylates the trans-cinnamic acid generated to para-coumaric acid. Furthermore, 4CL converts para-coumaric acid to para-coumaroyl-CoA, the only active form recognized by the ester-generating enzyme rosmarinic acid synthase, (RAS; EC 2.3.1.140), with the aid of CoA [11]. Although the expression of these genes has been studied in a variety of plant species, the RA biosynthesis pathway genes in Iranian lemon balm ecotypes, have yet to be evaluated. Different time frames, development stages, and reactions to biotic and abiotic stimuli impact the production and accumulation of these molecules [12]. Elicitors are one of the key variables that induce plant defensive responses, resulting in the build-up of targeted secondary metabolites [13,14,15]. The signal molecule methyl jasmonate (MeJA) plays a crucial function in the signal transduction pathway. It is a powerful abiotic elicitor that is used exogenously in plants to stimulate the formation of the desired secondary chemicals [16,17]. MeJA has been extensively employed to investigate the control of secondary metabolism in a variety of medicinal plants, including *Salvia miltiorrhiza* [18], *Satureja khuzistanica* [19], *Capparis spinosa* [20,21] and *Thymus migricus* [22]. MeJA has been shown to have a favourable effect on RA levels and the expression levels of associated critical genes in plants in many investigations [23,24,25].

Despite the existence of numerous studies examining the effect of elicitors, such as MeJA, on phenolic acid production, RA, and the expression pattern of RA-related genes in some species of the Lamiaceae family, have not, to our knowledge, been investigated in Iranian lemon balm ecotypes. Thus, the purpose here was to examine, for the first time, changes in the expression of important RA genes (*Mo**PAL*, *Mo**4CL*, and *Mo**RAS*), the accumulation of phenolic compounds, and the RA content of two Iranian lemon balm ecotypes (Esfahan and Ilam) in response to MeJA. In the other words, we sought to determine the concentrations of RA, which is an important compound found in lemon balm leaves after the plant was elicited with MeJA. We also attempted to establish a relationship between the pattern of expression of certain RA biosynthesis pathway genes and the changed content of RA in the leaves of lemon balm plants at the vegetative development stage after treating them with MeJA.

## 2. Result and Discussion

### 2.1. Changes in RA under Various MeJA Concentrations

The effect of various MeJA concentrations on RA, a main ingredient of lemon balm, was assessed (Figure 2). As shown in Figure 2, the RA content of both treated lemon balm ecotypes under MeJA elicitor increased significantly after 24 h. Generally, at all MeJA concentrations, RA accumulation was higher in the Esfahan ecotype than in the Ilam ecotype. The amount of RA in the Esfahan ecotype treated with 10, 100, 150, and 200 µM MeJA increased, which was 1.45-, 2.5-, 4.18-, and 3.47-fold higher than in the untreated plants. The corresponding use of these MeJA treatments enhanced the RA amount in the Ilam ecotype, which was 2.85-, 4.48-, 7.43-, and 5.72-fold more than their control (Figure 2). Therefore, RA amount differences of the two lemon balm ecotypes indicate that RA production process and amount are related to genotype-specific response to abiotic elicitors. In other words, our findings indicate that MeJA treatments may significantly increase the amount of RA present in different ecotypes of lemon balm, and that the gene expression patterns of RA biosynthesis-related genes are influenced by these elicitors. On the other hand, these findings suggest that RA levels in lemon balm plants vary depending on genotype and dose, which is in line with the findings of other researchers [20,21,22].

It was supposed that the high amount of RA observed after treatments of 100–150 µM MeJA was the optimum dose for the treatment of Lamiaceae families [26]. In accordance with these results, treatment of caper plants with 150 µM MeJA could induce flavonoid production [20]. Some researchers have noted that MeJA induces activities of the *PAL* enzyme, which may play a role in increasing RA in various plants [19,24]. According to the findings of the current study, the increase in the rutin amount under 100 µM to 150 µM treatment could be due to the influence of MeJA [20,21]. Our results demonstrate that the metabolic characteristics were noticeably influenced by the applied MeJA for both ecotypes. Finally, the rise in RA accumulation by MeJA stimuli could be due to the excitation of biosynthesis pathways and expression of related genes to activate radical scavenging through phenolic constituents. MeJA increased the amount of RA in both ecotypes, but the Esfahan ecotype showed a quicker effect.

Earlier findings corroborate our studies on the influence of various elicitors, especially MeJA, on the accumulation of RA and gene expression in lemon balm plant ecotypes. MeJA was shown to be the greatest elicitor for the formation of RA in *Coleus forskohlii* hairy root cultures, nodal culture of *Satureja khuzistanica* and *Agastache rugosa* Kuntze cell cultures, supporting this suggestion [26,27,28]. A biotic and abiotic elicitor enhanced the synthesis of RA in *A. rugosa* callus suspension cultures, according to Park et al. [29]. Kintzios et al. [30] previously observed that *Ocimum basilicum* cell suspension cultures accumulated RA up to 10 mg/g DW. Mizukami et al. [31] found a 10-fold rise in RA concentration in *Lithospermum erythrorhizon* cells treated with MeJA.

### 2.2. Effect of Various Concentrations of MeJA on Total Phenol Content (TPC) and Total Flavonoid Content (TFC)

Clearly, all concentrations of MeJA affect the TPC and TFC levels of Iranian lemon balm leaves (Figure 3). The pattern of TPC and TFC changes in both Esfahan and Ilam lemon balm ecotypes during the MeJA treatment were likely to reflect the pattern of RA content changes (Figure 2 and Figure 3A,B). Our results also showed that phenol and flavonoid accumulation were affected by MeJA, but with a different intensity, being higher in the MeJA-treated Esfahan ecotype than in the MeJA-treated Ilam ecotype. Our findings are consistent with other studies that show MeJA has been identified as a signalling molecule that promotes flavonoid accumulation in other plant species [32,33]. As a result of a rise in phenolic compounds, it is possible that the enhanced expression of these genes in response to MeJA is a result of this increase.

The 10, 100, 150, and 200 µM MeJA concentrations applied in this study enhanced TPC amount in the Esfahan ecotype, being 1.21-, 1.36-, 1.69-, and 1.52-fold higher than the untreated plants (Figure 3A). Corresponding TPC values were measured for the Ilam ecotype under the treatment of the different MeJA concentrations, being 1.38-, 1.85-, 2.72-, and 2.15-fold higher than the controls, respectively (Figure 3A). In addition, treatment of Esfahan lemon balm ecotypes with different concentrations of MeJA led to TFC of up to 6.73, 10.92, 21.51, and 14.57 mg QUE/g DW, respectively, these amounts being 1.79-, 2.91-, 5.75-, and 3.89-fold higher than the control plants (Figure 3B). Additionally, in the Ilam lemon balm ecotype, MeJA treatments increased the amount of TFC, which were approximately 2-, 2.96-, 5-, and 3.39-fold more than the content in non-treated plants (Figure 3B). These results show that TPC and TFC changes in lemon balm leaves vary depending on ecotype, which is in accordance with the results of previous researchers [34]. In addition, in both ecotypes considered, TPC values were higher than TFC values in untreated and all treated leaves, which is in line with earlier findings [35]. Our result is also consistent with other reports [36,37,38] stating that MeJA stimuli had a significant effect on TPC in different plants. Based on the same reports, MeJA, as a signalling molecule, has the potential to increase secondary metabolite accumulation in different plant species [39,40,41,42].

Jaafar et al. [43] proposed that increased polyphenolic compounds could be the outcome of increasing degradation of larger phenolic compounds to smaller compounds. Other researchers have also postulated that *PAL*, as the major regulatory enzyme involved in phenylpropanoid metabolism [40], may be associated with induction of TPC and TFC in MeJA-treated lemon balm ecotypes. Thus, the increase in RA and phenolic compound production by MeJA treatments was in accordance with the expression of RA biosynthetic pathway genes. The expressions of phenylpropanoid biosynthetic genes (*PAL*, *C4H*, and *4CL*) were reported to correlate with the flavonoid content in several plants [44,45].

### 2.3. Influence of MeJA on Expression MoPAL, Mo4CL, and MoRAS Genes

The qRT-PCR technique was utilized in this work to analyse changes in the transcript abundance of genes involved in the RA production pathway, as well as the association between gene expression and RA accumulation in Iranian lemon balm ecotypes treated with different MeJA concentrations (Figure 4). The results clearly show that the mRNA transcript levels of the three studied genes in lemon balm plants responded significantly to different MeJA treatments. It is worth noting that transcriptional levels of genes involved in RA biosynthesis were higher in MeJA-treated Esfahan ecotypes than in MeJA-treated Ilam ecotypes, which could explain why MeJA-treated plants produced more RA, especially in the first and last committed steps of RA biosynthesis and *MoPAL* and *MoRAS* expression. In detail, in the Esfahan lemon balm ecotype, the expression level of *MoPAL* rose from 5.28-fold at MeJA 10 µM to 8.69-fold at MeJA 150 µM and stayed practically the same at 7.53-fold at MeJA 200 µM compared with untreated plants (Figure 4A). The transcript level of *Mo4CL* increased considerably compared with non-treated plants at MeJA 10 µM and 100 µM (5.2- and 5.66-fold, respectively), steadily climbing to 6.17-fold at MeJA 150 µM, and then falling to 4.35-fold (Figure 4B). Additionally, *MoRAS* expression rose strongly to 7.74-fold at MeJA 100 µM, swiftly peaking at 16.6-fold with MeJA 150 µM, and then decreasing to 11.24-fold at the next concentration (MeJA 200 µM) (Figure 4C).

In the Ilam ecotype, the expression level of *MoPAL* rose significantly (3.24-fold) in the plants following treatment with MeJA 10 µM compared with the control plant, whereas no significant difference emerged between treatments with MeJA 10 and 100 µM. At MeJA 150 µM, *MoPAL* expression was dramatically stimulated, being 5.87-fold higher than in control plants (Figure 4A). After 24 h of treatment with MeJA 10 µM, there was a considerable rise in *Mo4CL* (2.23-fold), increasing further to 4.22-fold with MeJA 100 µM. *Mo4CL* expression reached the highest level at MeJA 100 µM (4.22-fold), thereafter declining (at MeJA 150 and 200 µM); these were nonetheless 3.33- and 2.23-fold higher than the untreated level (Figure 4B). The expression of *MoRAS* gene in MeJA-treated Ilam ecotype began to increase at MeJA 10 µM (3.24-fold), increased further at MeJA 100 µM (4.64-fold) and 150 µM (7.73-fold), then falling to 5.89-fold at MeJA 200 µM. The expression pattern of the RA biosynthesis pathway, including *MoPAL* and *MoRAS*, showed the same pattern for both ecotypes, as the transcript levels of all genes increased at all MeJA concentrations (Figure 4). As a result of MeJA treatment, the expression of the *MoPAL*, *Mo4CL*, and *MoRAS* genes increased, and their expression patterns were directly congruent with the RA accumulation pattern. Additionally, the present study demonstrated that spraying balm lemon ecotypes with high MeJA concentration (200 µM) led to a reduction in the studied gene expression compared with the other concentrations. These results are consistent with Kianersi et al. [20,21,22], who found that exogenously applied MeJA in high doses caused expression inhibition. The influence of external stimuli on secondary metabolite production will be useful in determining the maximal yield of secondary metabolites and in elucidating their biosynthetic pathway(s) [46,47,48,49,50].

In the two ecotypes studied here, MeJA treatments variably up-regulated *MoPAL*, and its expressions followed the same pattern as that of RA accumulation. The expression levels of RA biosynthesis-related genes in various organs of *Ocimum basilicum* cultivars and *Agastache rugosa* exhibited comparable findings [29,51]. Because *PAL* activity rose in Lamiaceae species before RA accumulated, it has been suggested that *PAL* is a crucial enzyme for entrance into the phenylpropanoid pathway [9,52]. Our findings revealed that *MoPAL* plays a critical role in RA biosynthesis. Depending on the stress and the plant species, the pace of transcript levels and gene induction may vary. The expression profile of *PAL* in *Salvia miltiorrhiza* was investigated using a variety of treatments, which resulted in up-regulation in response to stimuli [53,54].

Earlier research has demonstrated that MeJA elicitation increases the activity of *PAL*, a critical enzyme in the phenylpropanoid pathway [55]. Similar to the findings of the current study, a significant increase in the transcription level of the *PAL* gene, the first enzyme of the phenylpropanoid pathway, was detected in *Mentha spicata* plant treated with MeJA and other species [24,56]. Additionally, in another study, after 24 h of treatment with MeJA, *PAL* gene expression and its enzymatic activity increased in basil such that the change in *PAL* gene expression at different time points was consistent with the accumulation of phenylpropanoid compounds [48]. Although the importance of *MoPAL* and *MoRAS* expression in RA biosynthesis in both ecotypes is obvious based on our findings, the difference in expression and RA accumulation quantities indicates that they are most likely associated with lemon balm ecotypes.

Our findings showed that *MoRAS* expression is tightly linked to RA biosynthesis. In both MeJA-treated Iranian lemon balm ecotypes, the expression pattern of *MoRAS* matched the rise in RA (Figure 2 and Figure 4). For example, expression of *MoRAS* at MeJA 150 µM after 24 h was 16.6- and 7.73-fold higher than in control plants in the Esfahan and Ilam ecotypes, respectively. In parallel, at this concentration, the RA contents were 4.18- (Esfahan ecotype) and 1.95-fold (Ilam ecotype) higher than in normal plants. In the ecotypes studied, the diminished production of RA, phenol, and flavonoid contents at MeJA 200 µM compared with MeJA 150 µM coincided with a lower *MoRAS*. The crucial involvement of this gene in the control of RA biosynthesis may be due to similar patterns of *MoRAS* expression and RA content.

The findings of this work are consistent with those of Kim et al. [25], who noted that MeJA treatment raised the transcript levels of phenylpropanoid biosynthesis genes *ArPAL*, *Ar4CL*, and *ArC4H* in *A. rugosa* cell suspension cultures, resulting in higher RA accumulation [25]. Elicitation with MeJA in plants induced *PAL* activity rapidly, according to Mizukami et al. [31]. Many reports have shown that MeJA affects the transcript levels of secondary metabolite biosynthetic pathway genes and causes the accumulation of bioactive compounds in various plant species [19,20,21,46,47,57]. Belhadj et al. [58] discovered that the expression of the *PAL* gene rose quickly in MeJA-treated grape leaves. Exogenous administration of MeJA, according to Farooq et al. [59], further altered the activity of *PAL*, *PPO*, and *CAD*, as well as their relative mRNA levels.

Surprisingly, the highest levels of *MoPAL* and *MoRAS* expression were identified at MeJA 150 µM, whereas the lowest levels were seen in control plants. This indicates that when the quantity of MeJA in a sample rises, the amount of RA in the sample rises as well. In the Ilam ecotype, however, leaves treated with MeJA 100 µM produced the greatest amounts of the *Mo4CL* transcript compared with the other doses. Finally, our findings reveal that MeJA treatments had a significant impact on RA content, total phenolic content (TPC), and total flavonoid content (TFC), as well as the expression of RA biosynthesis-related key genes (PAL, 4CL, and RAS) in Iranian lemon leaves.

## 3. Materials and Methods

### 3.1. Plant Material and Growth Conditions

The seeds of two Iranian lemon balm (*M. officinalis*) ecotypes (Esfahan and Ilam) were provided by the Pakan-Bazr Co. of Esfahan, Iran. The seeds were planted in pots (containing of a perlite-compost mixture) and then grown in a greenhouse under controlled light (16 h day/8 h night and a photosynthetic photon flux density of 290 μmol m^−2^ s^−1^) and temperature (25/19 °C day/night) conditions.

### 3.2. Application of MeJA treatments

In this study, the potted two-month-old *M. officinalis* plants at vegetative development stage (when the plants had only root, stem and leave parts) were treated with 10, 100, 150, and 200 μM MeJA and distilled water (as a control). In detail, MeJA (SIGMA-ALDRICH) solutions were filter-sterilized with a filter membrane (0.22 μm, MILLIPORE). After that, final concentrations of MeJA solutions (10, 100, 150, and 200 μM) and distilled water (control) were sprayed on the aerial parts of three lemon balm plants until runoff (1000 mL/treatment). For each treatment (MeJA and distilled water), three plants per replicate were considered and collected. After one day of the initial treatment, treated and untreated leaves were collected, frozen in liquid nitrogen, and kept at −80 °C until future use.

### 3.3. RNA Isolation, Synthesis of cDNA, and q-PCR Analysis

The total RNA of leaves of lemon balm ecotypes (100 mg of the frozen fresh leaves) were isolated based on the method described by manufacturer’s (SinaClon Bioscience Co., Karaj, Iran) guideline, as well as cDNA was synthesized using the 2-steps RT-PCR Kit (SinaClon Bioscience Co., Karaj, Iran) according to the manufacturer’s protocol [20]. By using the gene-specific primers and β-Actin gene (as a housekeeping gene), as previously reported [60] (Table 1), the impact of various concentrations of MeJA on mRNA transcript levels of *MoPAL*, *Mo4CL*, and *MoRAS* was analysed based on the fold-change (2^−ΔΔCt^) method, as described elsewhere [61]. Additionally, three replicates of biological and technical was used to gene expression analysis.

### 3.4. Rosmarinic Acid (RA) Extraction and HPLC Analysis

The method of Wang et al. [62] with slight modifications was used to evaluate the effects of various concentrations of MeJA on the accumulation RA. Briefly, 80 mg of the ground dried leaves were added to 50% ethanol (20 mL), and the solution was sonicated for 15 min and then centrifuged at 3000 rpm for 15 min. The reaction volume of 20 mL was achieved by adding sterile water to the supernatant. A 0.2 μM syringe filter was used for the resulting solution filtering before injection into an Agilent Technologies 1100 series HPLC system (C18 Column (250 × 4.6 mm)) in order to separate the RA.

Different concentrations of RA standard were prepared in 1 mL methanol/water (50/50 *v*/*v*) ranging from 1 to 300 mg/L. The peak areas obtained from the injections were used to calculate the calibration curve. The flow rate was 1 mL min^−1^ and included mobile phase eluent A (water/1% H3PO4) and eluent B (methanol/1% H3PO4). Finally, after detection of the RA compound at a wavelength of 333 nm, the RA concentration was stated as mg/g dry weight. All samples were analysed in triplicate. The chromatography peak of RA was confirmed according to the retention time of the reference standard. The quantitative analysis was performed with external standardisation by measurement of the peak areas using Agilent ChemStation software.

### 3.5. Assay of Total Phenolic and Flavonoid Contents

To determine the total phenolic content (TPC), methanolic extracts of 1000 mg of dried and crushed fennel leaf samples were prepared by suspending them in 80% methanol (25 mL) and shaking for 24 h at room temperature on a shaker (150 rpm). The extracts were then filtered through two Whatman paper, and TPC was determined using the Folin–Ciocalteu reagent, as previously reported [34]. To begin, 0.5 mL of each sample’s methanolic extract was mixed well with 2.5 mL of Folin–Ciocalteu reagent (10-fold diluted) and 2 mL of sodium carbonate (7.5%). Finally, after 15 min of heating at 45 °C, absorbance at 765 nm was determined, and TPC was calculated as mg tannic acid equivalent/g dry weight (DW).

Additionally, the total flavonoid content (TFC) was investigated using the aluminium chloride colorimetric method described by Zhang et al. [63]. First, 0.25 mg of each sample extract was combined with 1.25 mL of water and 0.75 mL of sodium nitrate and incubated for 6 min in the dark. To finish the reaction, 0.15 mL of aluminium chloride (10%) was added to the mixture and incubated in the dark for 300 s. Finally, 0.275 mL of water and 0.5 mL of sodium hydroxide solution were added to each sample (5%). After reading the adsorption of the reaction solution at 510 nm, the TFC was presented as mg quercetin equivalent/g DW.

### 3.6. Statistical Analysis

The RA content, TPC, TFC and gene expression of two Iranian ecotypes of lemon balm treated with different concentrations of MeJA were analysed by using factorial experiments based on the completely randomized design (CRD). Three replications were used in each experiment. SPSS 16 statistical software was used to perform ANOVA on all data. The means were compared using Duncan’s multiple range test (DMRT).

## 4. Conclusions

For the first time, we demonstrated that the exogenous application of MeJA in Iranian lemon balm ecotypes can increase the amount of RA, TPC, and TFC, as well as the transcript levels of important genes involved in the RA (phenylpropanoid) pathway, including *MoPAL*, *Mo4CL*, and *MoRAS*. More research is required to investigate the expression of other genes implicated in this pathway and their interaction with phenolic compound accumulation under MeJA, and other treatments that include abiotic stress factors. Future studies may enable genetic manipulation of this pathway to enhance the production of the valuable compounds in lemon balm.

## Figures and Tables

**Figure 1 molecules-27-01715-f001:**
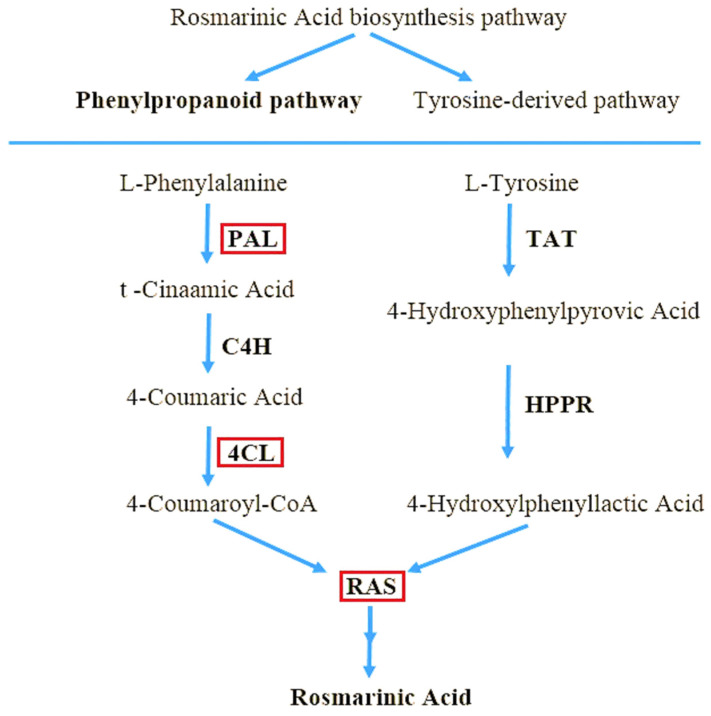
A rectangle is placed above enzyme genes that have been investigated in Iranian lemon balm including Phenylalanine ammonia-lyase (PAL), 4-Coumarate-CoA ligase (4CL), Rosmarinic acid synthase (RAS).

**Figure 2 molecules-27-01715-f002:**
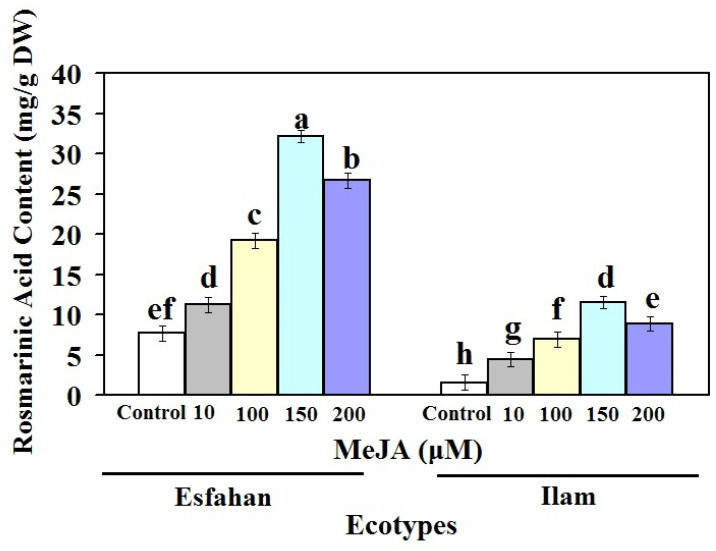
Rosmarinic acid content of leaves of Iranian lemon balm ecotypes treated with different treatments of MeJA. Bars with different letters mean significant at 1% level of probability according to Duncan’s test.

**Figure 3 molecules-27-01715-f003:**
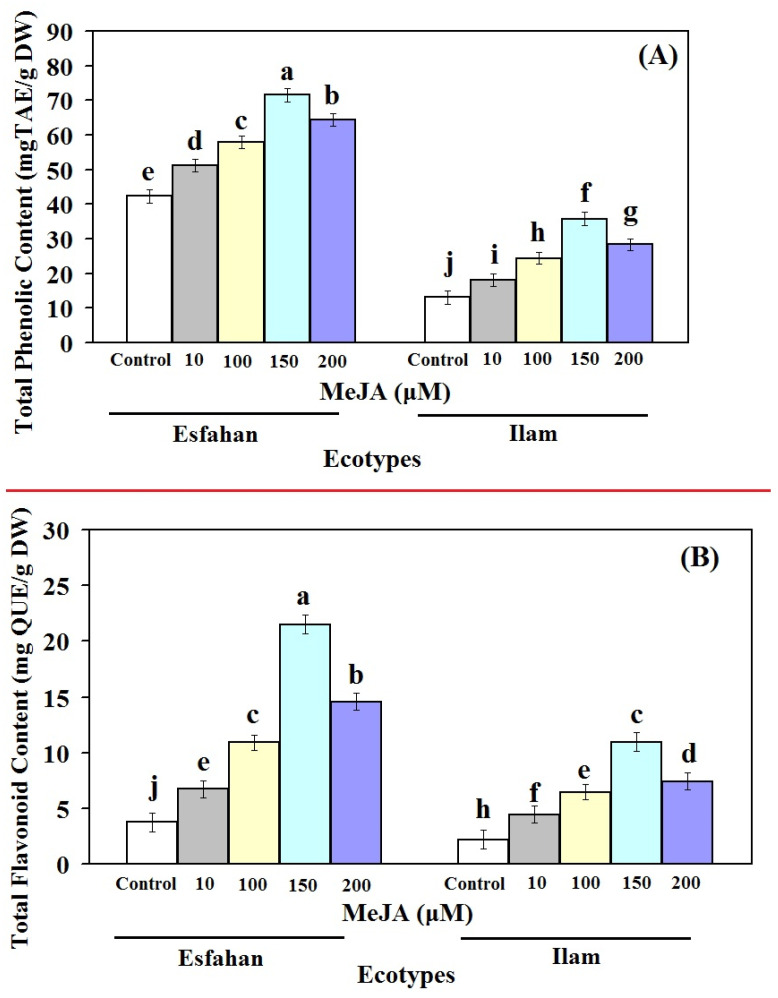
Effect of different concentrations of MeJA on TPC (**A**) and TFC (**B**) of leaf extracts of two Iranian lemon balm ecotypes. The results are expressed as means ± SD (*n* = 3). In each column different letters mean significant at 1% level of probability according to Duncan’s test.

**Figure 4 molecules-27-01715-f004:**
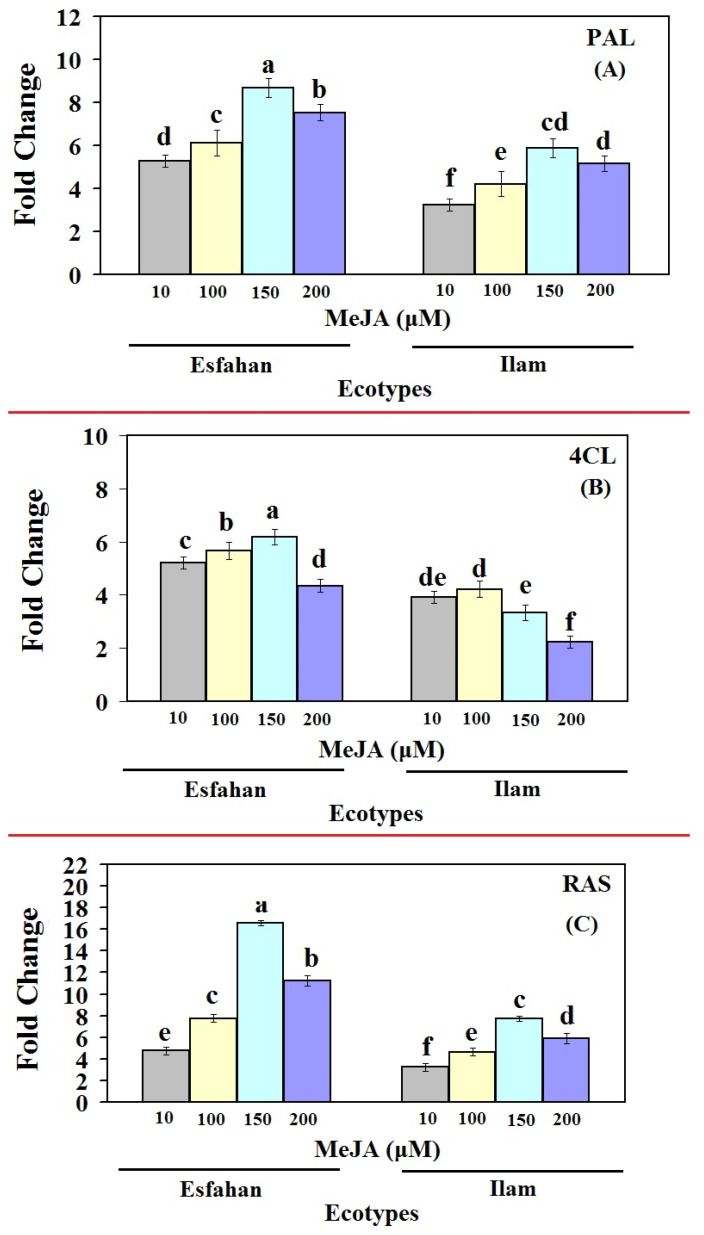
Gene expression (Fold-change) of first, middle and late pathway genes of rosmarinic acid biosynthesis pathway, *PAL* (**A**), *4CL* (**B**) and *RAS* (**C**), in leaves of control (untreated) and MeJA-treated lemon balm plants. qRT-PCR was based on the Ct values. The Ct value for each sample was normalized using the reference gene *β-Actin*. Bars with different letters are significantly (*p* ≤ 0.01) different according to Duncan’s test. Error bars indicate standard error values.

**Table 1 molecules-27-01715-t001:** Primers used to qRT-PCR analysis.

Real-Time Primers	Sequences (5ʹ to 3ʹ)
PAL *F* PAL *R*	CCGAAGTCATGAACGGAAAGC CGCAGCCTTAACATAACCGCTC
4CL *F* 4CL *R*	AGACGATCATGCTCTTGCTCCC GGCCTTGGCTTGCTTGATTACC
RAS *F* RAS *R*	ACGCCCCGACCTCAACCTTATC AAGTGGTGCTCGTTTGCCACG
β-Actin β-Actin	TGTATGTTGCCATCCAGGCCG AGCATGGGGAAGCGCATAACC

## Data Availability

Not applicable.

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
