# Peer review of "Change in Secondary Metabolites and Expression Pattern of Key Rosmarinic Acid Related Genes in Iranian Lemon Balm (Melissa officinalis L.) Ecotypes Using Methyl Jasmonate Treatments"

_molecules, 2022, doi:10.3390/molecules27051715_

Round 1

Reviewer 1 Report

Kianersi et al. describes the elicitation RA, phenolic, and flavonoid metabolites and genes by MeJA. The elicitation experiments carried out in this study were sound but the analytical methods left a few questions for this reviewer. Moreover, while this study could be of interest, its area of focus has been described in many other studies using different plant species; therefore, the novelty of this work came into question.

  1. The data in this study shows that MeJA elicits an RA, phenolic, and flavonoid related metabolic and genetic response in Iranian lemon balm, but as the authors mention, this has been shown in many other plant species. Therefore, the novelty of this study is unclear. Perhaps additional experiments or comparisons could be performed to add novelty to the overall study.   
  2. Confidence in the analytical approaches would be higher if the authors provided some additional information and explanation in the methods. In the HPLC analysis of RA, was an RA standard used to confirm the absorbance and retention time of RA? Including chromatograms in supplemental material of the commercial or purified standard (RA) and the biological sample would be helpful. In the measurement of phenolic acids, don’t many phenolic compounds have absorbance between 200-360 nm? Does the Folin-Ciocalteu reagent change the absorbance of phenolics? Clarification surrounding this should be provided in the methods section. Without standards, it is also difficult to know whether other non-phenolic acid molecules with absorbance at 765 nm are being included in the detection and analysis. The same is true for the flavonoid analysis. How do the authors account for non-flavonoids that could absorb at 510 nm?
  3. The manuscript could use some polishing for content and English errors. For example, lines 156-157 should read that “….MeJA can differentially influence phenol and flavonoid accumulation.” The manuscript should be reviewed and corrections made throughout. 

Author Response

Dear reviewer,

Thank you so much for your consideration and giving us an opportunity to revise the paper. The your opinions were very valuable that caused to improved the quality and quantity of our article. 
We hope you like it and find it suitable for publication. I provided the proper answer (point to point) which is addressed here and pink (First reviewer) highlighted parts in the text, too
Sincerely,
Farzad Kianersi

Reviewer 1.

Kianersi et al. describes the elicitation RA, phenolic, and flavonoid metabolites and genes by MeJA. The elicitation experiments carried out in this study were sound but the analytical methods left a few questions for this reviewer. Moreover, while this study could be of interest, its area of focus has been described in many other studies using different plant species; therefore, the novelty of this work came into question.

Thank you for your positive opinion and interesting.

1-The data in this study shows that MeJA elicits an RA, phenolic, and flavonoid related metabolic and genetic response in Iranian lemon balm, but as the authors mention, this has been shown in many other plant species. Therefore, the novelty of this study is unclear. Perhaps additional experiments or comparisons could be performed to add novelty to the overall study.   

Response: Thank you for your consideration. We used the results of previous findings, because earlier findings corroborate and support our studies on the influence of MeJA elicitors on accumulation of RA and genes expression in lemon balm plant ecotypes as well as to date, none scientific information have been reported on the expression of the genes for the RA biosynthetic pathway for Iranian lemon balm species or ecotypes.

On the other hand, there is some information on Lemon balm obtained and an articles related to the lemon balm transcriptome were published in Journal of Plant Physiology (2014) and Annual Research & Review in Biology (2014), but the authors did not expression the genes for the RA biosynthesis pathway under different concentration of MeJa in Iranian lemon balm and so far no report has been published. Also, the words of "first report or for the first time" mentioned in different section of manuscript that indicate the novelty of our study as clearly.

On the other hand, with respect to the referee, as you know, additional experiments require more time and in present project we are not left with enough time for adding these experiments, and it is not possible at a moment because we do not access to plant material and Covid-19 situation. So unfortunately we would not be able to add these additional experiments in current manuscript, but we will do additional experiments in our next planned projects.

2- Confidence in the analytical approaches would be higher if the authors provided some additional information and explanation in the methods. In the HPLC analysis of RA, was an RA standard used to confirm the absorbance and retention time of RA? Including chromatograms in supplemental material of the commercial or purified standard (RA) and the biological sample would be helpful.

Response: The complementarity information were inserted in the methods section.

In the measurement of phenolic acids, don’t many phenolic compounds have absorbance between 200-360 nm? Does the Folin-Ciocalteu reagent change the absorbance of phenolics? Clarification surrounding this should be provided in the methods section. Without standards, it is also difficult to know whether other non-phenolic acid molecules with absorbance at 765 nm are being included in the detection and analysis. The same is true for the flavonoid analysis. How do the authors account for non-flavonoids that could absorb at 510 nm?

Response: With respect to the referee, as explained in polyphenolic compounds analysis section, we thoroughly and accurately done all biochemical and statistical analyzes required to interpret the experiment data. Also, there are different and important manuscripts about the way and method of polyphenolic compounds (TPC and TFC) in various plants (Salami et al, 2014; Fooladi vanda et al., 2019, Fatemi et al., 2019, 2020; Kianersi et al. 2021; Abdollahi et al., 2022), which had previously shown that the that the these methods was considered.

Yes, Folin-Ciocalteu′s phenol reagent does not contain phenol. Rather, the reagent will react with phenols and non-phenolic reducing substances to form chromogens that can be detected spectrophotometrically.

On the other words, as other investigators have suggested, the Fiolin-Ciocalteu assay should be seen as a measure of total antioxidant capacity rather than phenolic content but due to phenolics are the most abundant antioxidants in most plants, it gives a rough approximation of total phenolic content in most cases (Everette et al., 2010).

On the other hand, the Salami et al., (2016) method (with 35 citation) that has applied specifically for fennel, was used to assay of phenolic compound in this study.

Therefore, Total phenolic contents and Total flavonoid contents of leaf samples were assessed using a standard curve achieved with standards of tannic acid and quercetin (Sigma-Aldrich, Germany), respectively.

3- The manuscript could use some polishing for content and English errors. For example, lines 156-157 should read that “….MeJA can differentially influence phenol and flavonoid accumulation.” The manuscript should be reviewed and corrections made throughout.

Response: Corrections were inserted into the text. 

The all of sections and English language revised by Dr. Peter Poczai (native person) as co-author and corresponding author of manuscript.

Reviewer 2 Report

Whole Manuscript

Try to improve language and use small sentence with clear meaning

Line 20

Iranian lemon balm ecotypes, as first report,. “Remove (,)”

Line 28

Use appropriate world despite of “discovery

Line 39 and whole manuscript

M. officinalis (use italics font)

Whole manuscript

Try to shorten the sentences

Line 96

MeJA  instead of “MeJa”

Results and discussion

Submit diagram in some other form, should be more clear

Line 325

RNA isolation kit (Name of manufacturer)

Line 350

Mention the number of Whatman paper

References

Check the style of reference (mismatch somewhere)

Author Response

Dear reviewer,

Thank you so much for your consideration and giving us an opportunity to revise the paper. The your opinions were very valuable that caused to improved the quality and quantity of our article. 
We hope you like it and find it suitable for publication.
I provided the proper answer (point to point) which is addressed here and yellow (Second reviewer) highlighted parts in the text, too.

Sincerely,
Farzad Kianersi

Reviewer 2.

1-Whole Manuscript: Try to improve language and use small sentence with clear meaning.

Response: The all of sections and English language revised by Dr. Peter Poczai (Native person) as co-author and corresponding author of manuscript.

2- Line 20: Iranian lemon balm ecotypes, as first report, “Remove (,)”.

Response: Corrections were inserted into the text.

3- Line 28:  Use appropriate world despite of “discovery”.

Response: Corrections were inserted into the text.

4- Line 39 and whole manuscript:   M. officinalis (use italics font)

Response: Corrections were inserted into the text.

5- Whole manuscript:  Try to shorten the sentences

Response: On the recommendation of the honorable reviewer, several unnecessary sentences were removed, and the sentences were shorten and improved.

6- Line 96:       MeJA instead of “MeJa”.

Response: Corrections were inserted into the text.

7- Results and discussion: Submit diagram in some other form, should be more clear.

Response: The new figures (2, 3 and 4) with the high quality inserted in the manuscript file.

8- Line 325: RNA isolation kit (Name of manufacturer).

Response: Corrections were inserted into the text. The name of manufacture added to the text.

9- Line 350: Mention the number of Whatman paper.

Response: Corrections were inserted into the text.

10- References:  Check the style of reference (mismatch somewhere).

On the recommendation of the honorable reviewer, several references were improved.

Reviewer 3 Report

This manuscript entitled “Changes in Secondary Metabolites and Expression Pattern of Key Rosmarinic Acid Related Genes in Iranian Lemon Balm (Melissa officinalis L.) Ecotypes using Methyl Jasmonate treatments” (molecules-1605298) was studied the increase in Rosmaric acids contents, total phenolic contents, and total flavonoid contents respond to methyl jasmonic acid in two ecotype of lemon balm.

And then, the authors measured the changes in gene expression of phenylpropanoid synthesis pathways related genes (PAL, 4CL, and RAS) in MeJA-treated lemon balm plants.

There are few detailed studies about MeJA responses in non-model plants. Thus I feel this study is meaningful to increase in the knowledges of changes in metabolism of this species with MeJA, and useful to increase in production of functional metabolites in the medicinal herb plants.However, I think there are some issues in this manuscript. 

Major point

  • The authors use two ecotypes of lemon balm (Eshahan and Ilam) and compare the measurements (RA content, TPC, TFC, and gene expression changes) between the two ecotypes. However, it does not say why these differences occurred. This point may be significant in this study. It needs to be clarified why such differences occurred in the manuscript. 
  • The "results and discussion" section seem to indicate too much that the results are similar to those of previous studies. If this is not done, the novelty of this study may not be conveyed to the reader. The author should describe the novelty of this study more clearly.  

Minor points

  • All gene names and scientific names of plants should be in italics.
  • p.3 line 101 and 104, p.5 line 159 and 161; measured values were already indicated as graphs (Fig. 2 and Fig. 3). These values may not be considered necessary in the text.-p.8 line 316-317; culture conditions (light intensity, etc.) should be shown in detail.
  •  p.8 line 319-320; It would be necessary to explain how MeJA is treated to lemon balm (spray or drip, how many doses per plant) and the condition of the two-month-old plants (when MeJA is treated).
  • p.9, line 324; the materials (lemon balm leaves for RNA extraction, i.e. fresh weight and number of sheets) need to be described.-The concentration of MeJA in the legend of the graph should be listed below the horizontal line as follows.

Control 10 100 150 200         

MeJA (µM)

  • -The entire manuscript should be checked by an English editing.

Author Response

Dear reviewer,

Thank you so much for your consideration and giving us an opportunity to revise the paper. The your opinions were very valuable that caused to improved the quality and quantity of our article. 
We hope you like it and find it suitable for publication.
I provided the proper answer (point to point) which is addressed here and green (Third reviewer) highlighted parts in the text, too.

Sincerely,
Farzad Kianersi

Reviewer 3.

Comments and Suggestions for Authors

This manuscript entitled “Changes in Secondary Metabolites and Expression Pattern of Key Rosmarinic Acid Related Genes in Iranian Lemon Balm (Melissa officinalis L.) Ecotypes using Methyl Jasmonate treatments” (molecules-1605298) was studied the increase in Rosmaric acids contents, total phenolic contents, and total flavonoid contents respond to methyl jasmonic acid in two ecotype of lemon balm.

And then, the authors measured the changes in gene expression of phenylpropanoid synthesis pathways related genes (PAL, 4CL, and RAS) in MeJA-treated lemon balm plants.

There are few detailed studies about MeJA responses in non-model plants. Thus I feel this study is meaningful to increase in the knowledges of changes in metabolism of this species with MeJA, and useful to increase in production of functional metabolites in the medicinal herb plants.However, I think there are some issues in this manuscript.

Response: Thank you for your consideration, interesting and your suggestion.

Major point

1- The authors use two ecotypes of lemon balm (Eshahan and Ilam) and compare the measurements (RA content, TPC, TFC, and gene expression changes) between the two ecotypes. However, it does not say why these differences occurred. This point may be significant in this study. It needs to be clarified why such differences occurred in the manuscript. 

Response: Corrections were inserted into the text. The "goal of research" added into the introduction section as following: we sought to determine the concentrations of RA, which is an important compound found in two Iranian lemon balm leaves after the plant was elicited with MeJA. We also attempted to establish a relationship between the pattern of expression of certain RA biosynthesis pathway genes and the changed content of RA in the leaves of lemon balm plants at the vegetative development stage after treating them with MeJA.

On the other hand, these findings suggest that differences for all aspects (TPC, TFC, RA and Gene expression) in two ecotypes of lemon balm plants vary depending on genotype and dose, which is in line with findings of other researchers (Salami et al. 2016; Kianersi et al., 2021; Abdollahi et al., 2022).

2- The "results and discussion" section seem to indicate too much that the results are similar to those of previous studies. If this is not done, the novelty of this study may not be conveyed to the reader. The author should describe the novelty of this study more clearly.

Response: On the recommendation of the honorable reviewer, several unnecessary sentences were removed, and the sentences were shorten and improved.

However, we used the results of previous studies and findings, because earlier findings corroborate and support our studies on the influence of MeJA elicitors on accumulation of RA and genes expression in lemon balm plant ecotypes.

Our study and research is novel. As we explained in introduction and results and discussion, to date, none scientific information have been reported on the expression of the genes for the RA biosynthetic pathway for Iranian lemon balm species or ecotypes. On the other hand, there is some information on Lemon balm obtained and an articles related to the lemon balm transcriptome were published in Journal of Plant Physiology (2014) and Annual Research & Review in Biology (2014), but the authors did not expression the genes for the RA biosynthesis pathway under different concentration of MeJa in Iranian lemon balm and so far no report has been published. Also, the words of "first report or for the first time" mentioned in different section of manuscript that indicate the novelty of our study as clearly.

Minor points

3- All gene names and scientific names of plants should be in italics.

Response: Corrections were inserted into the text.

4- p.3 line 101 and 104, p.5 line 159 and 161; measured values were already indicated as graphs (Fig. 2 and Fig. 3). These values may not be considered necessary in the text.

Response: Thank you for your opinion. Corrections were inserted into the text.

5- p.8 line 316-317; culture conditions (light intensity, etc.) should be shown in detail.

Response:  Corrections were inserted into the text.

6- p.8 line 319-320; It would be necessary to explain how MeJA is treated to lemon balm (spray or drip, how many doses per plant) and the condition of the two-month-old plants (when MeJA is treated).

Response:  Corrections were inserted into the text. The additional information were inserted in the methods section.

7- p.9, line 324; the materials (lemon balm leaves for RNA extraction, i.e. fresh weight and number of sheets) need to be described.

Response: Corrections were inserted into the text. The weights of the samples used in the molecular analysis were added to the text. However, in principle, the number of sheets, etc do not write in section of molecular analysis, because each method requires its own sample size, number and weight (Fatemi et al., 2019; Kianersi et al., 2020 a,b; Kianersi et al, 2021; Abdollahi et al., 2022).

8- The concentration of MeJA in the legend of the graph should be listed below the horizontal line as follows.

Control 10 100 150 200

MeJA (µM)

Response: Corrections were inserted into the text. The new figures (2, 3 and 4) with the high quality inserted in the manuscript file.

9- The entire manuscript should be checked by an English editing.

Response: The all of sections and English language revised by Dr. Peter Poczai (Native person) as co-author and corresponding author of manuscript.

Round 2

Reviewer 1 Report

The authors have addressed concerns 2 and 3, but novelty is still not abundantly obvious in this reviewers opinion.